# GWAS in childhood acute lymphoblastic leukemia reveals novel genetic associations at chromosomes 17q12 and 8q24.21

Joseph L. Wiemels[1,2,3,4], Kyle M. Walsh[1,2], Adam J. de Smith[1], Catherine Metayer[5], Semira Gonseth [1,4], Helen M. Hansen[2], Stephen S. Francis[1,6], Juhi Ojha[1], Ivan Smirnov[2], Lisa Barcellos[5], Xiaorong Xiao[5], Libby Morimoto[5], Roberta McKean-Cowdin[4], Rong Wang[7], Herbert Yu [8], Josephine Hoh[7], Andrew T. DeWan[7] & Xiaomei Ma[7]

Childhood acute lymphoblastic leukemia (ALL) (age 0–14 years) is 20% more common in Latino Americans than non-Latino whites. We conduct a genome-wide association study in a large sample of 3263 Californian children with ALL (including 1949 of Latino heritage) and 3506 controls matched on month and year of birth, sex, and ethnicity, and an additional 12,471 controls from the Kaiser Resource for Genetic Epidemiology Research on Aging Cohort. Replication of the strongest genetic associations is performed in two independent datasets from the Children's Oncology Group and the California Childhood Leukemia Study. Here we identify new risk loci on 17q12 near *IKZF3/ZPBP2/GSDMB/ORMDL3*, a locus encompassing a transcription factor important for lymphocyte development (*IKZF3*), and at an 8q24 region known for structural contacts with the *MYC* oncogene. These new risk loci may impact gene expression via local (four 17q12 genes) or long-range (8q24) interactions, affecting function of well-characterized hematopoietic and growth-regulation pathways.

[1] Department of Epidemiology and Biostatistics, University of California San Francisco, 1450 3rd Street, San Francisco, CA 94158, USA. [2] Department of Neurological Surgery, University of California San Francisco, 1450 3rd Street, San Francisco, CA 94158, USA. [3] Institute for Human Genetics, University of California San Francisco, 1450 3rd Street, San Francisco, CA 94158, USA. [4] Department of Preventative Medicine, University of Southern California, SSB 318D 2001 N. Soto Street, Los Angeles, CA 90033, USA. [5] School of Public Health, University of California Berkeley, 1950 University Avenue, Suite 460, Berkeley, CA 94720, USA. [6] Department of Epidemiology, School of Community Health Sciences, University of Nevada Reno, 1664 N. Virginia Street, Reno, NV 89557, USA. [7] Department of Chronic Diseases Epidemiology, School of Public Health, Yale University, 60 College Street, New Haven, CT 06520, USA. [8] University of Hawaii Cancer Center, 701 Ilalo Street, Honolulu, HI 96813, USA. Joseph L. Wiemels and Kyle M. Walsh contributed equally to this work. Andrew T. DeWan and Xiaomei Ma jointly supervised this work. Correspondence and requests for materials should be addressed to J.L.W. (email: joe.wiemels@ucsf.edu) or to X.M. (email: xiaomei.ma@yale.edu)

Acute lymphoblastic leukemia (ALL), the most common type of cancer in children, has an increasingly clear genetic etiology with a role for both rare high-penetrance genetic predisposition alleles[1] and more common low-penetrance alleles. These latter alleles were discovered via genome-wide association studies (GWAS), which identified genetic risk factors at *ARID5B*, *CEBPE*, *IKZF1*, *CDKN2A*, *PIP4K2A*, and *GATA3*[2–5]. Additional loci were recently identified at *LHPP* and *ELK3*[6]. While most of these genes encode hematopoietic transcription factors (TFs), the causal variants underlying these genetic associations remain unclear, apart from a missense polymorphism in *CDKN2A*[7–9] and single-nucleotide polymorphisms (SNPs) disrupting TF-binding sites in the promoters of *CEBPE*[10] and *ARID5B*[11]. These prior GWAS focused on non-Latino white populations; however, the highest rate of childhood ALL globally is found in Latinos in the United States (US)[12]. We sought to identify additional risk alleles involved in the etiology of ALL in both Latinos and non-Latinos by leveraging a large, ethnically diverse population in California in conjunction with an array specifically designed for the Latino population, while supplementing with additional accessible controls genotyped on the same platform. We replicated our findings in two independent childhood ALL case–control datasets from the Children's Oncology Group (COG) and the California Childhood Leukemia Study (CCLS). We identify two new genetic loci impacting childhood ALL risk, which will aid in our understanding of the etiology of this disease.

## Results

**Association analysis**. After quality control, 757,935 polymorphic autosomal SNPs and 19,240 subjects (3263 cases, 15,977 controls, which included genotyped controls run specifically for this study and additional public controls, see Supplementary Table 1) remained for discovery analyses. Case–control association tests were stratified by race/ethnicity including 10,533 Latino subjects (1949 cases, 8584 controls), 4735 non-Latino white subjects (1184 cases, 3551 controls), and 3972 African-American subjects (130 cases, 3842 controls), and then the race/ethnicity-stratified GWAS were combined via meta-analysis. Sample size and ancestral heterogeneity precluded the inclusion of more minor

populations (e.g., Southeast Asians, Indian subcontinental, and other races/ethnicities) for the current analysis. Associations reached genome-wide statistical significance (i.e., $P < 5.0 \times 10^{-8}$ in an additive logistic regression analysis) at known ALL risk loci, including: *ARID5B*, *IKZF1*, *PIP4K2A*, and *CEBPE* among Latinos, and *ARID5B*, *IKZF1*, and *PIP4K2A* for non-Latino whites (Supplementary Table 2). No SNPs reached genome-wide significance among African Americans, likely due to the relatively small sample size. Analyses by each racial/ethnic subgroup did not reveal any novel associations at genome-wide statistical significance.

The three race/ethnicity-stratified discovery GWAS revealed associations reaching genome-wide statistical significance (i.e., $P < 5.0 \times 10^{-8}$ using a fixed-effects meta-analysis) at five known ALL risk loci (*ARID5B*, *IKZF1*, *PIP4K2A*, *CEBPE*, and *CDKN2A*) (Fig. 1 and Supplementary Table 2) and three novel loci at chromosome 7p15.3 (rs2390536, near *SP4*, odds ratio (OR) = 1.20, 95% confidence interval (CI): 1.13–1.29, $P = 3.6 \times 10^{-8}$), chromosome 8q24.21 (rs4617118, in a gene desert, OR = 1.27, 95% CI: 1.17–1.38, $P = 3.1 \times 10^{-9}$), and chromosome 17q12 (rs2290400, near the hematopoietic TF *IKZF3*, OR = 1.18, 95% CI: 1.11–1.25, $P = 2.1 \times 10^{-8}$) (Fig. 1 and Table 1). For these novel loci, no heterogeneity in association was noted across the three ethnicities (heterogeneity $P$ values were insignificant and $I^2$ values were low, Supplementary Table 3). In addition, a random-effects meta-analysis applied to these data yielded similar ORs and $P$ values. Recently identified GWAS hits at *LHPP*, *ELK3*, and *GATA3* (rs35837782, rs4762284, and rs3824662, respectively)[3,6] were not directly genotyped on the Axiom array, but when those SNPs were imputed in our dataset, replication was achieved in a fixed-effect meta-analysis of the three race/ethnicity-stratified discovery GWAS ($P = 5.7 \times 10^{-6}$, $2.1 \times 10^{-3}$, and $2.3 \times 10^{-10}$, respectively, Supplementary Table 2).

Using 1000 Genomes SNP data as a reference[13], we imputed additional SNPs within a 1 Mb region centered on the three novel SNP associations. In parallel, we used additional genome-wide SNP array data to impute these regions in non-overlapping European-ancestry ALL patients from the COG ($N = 959$ cases) and controls from the Wellcome Trust Case–Control Consortium ($N = 2624$ controls), and 530 additional Latino ALL cases and 511

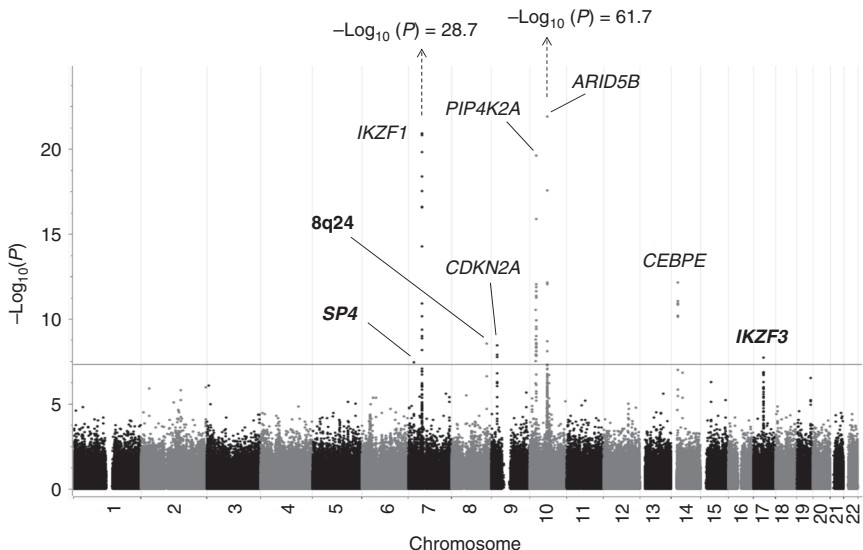

**Fig. 1** Manhattan plot of $P$ values resulting from meta-analysis of three separate analyses in the California population (Latinos, non-Latino whites, and non-Latino blacks) including study-specific controls and GERA controls (total $N = 3263$ cases, 15,977 controls). The locations of prior identified GWAS hits are indicated by their gene acronyms. New genetic associations identified in this study are highlighted in bold. These association peaks exceed the Bonferroni genome-wide $P$ value cutoff of $5 \times 10^{-8}$ indicated by a horizontal line

**Table 1 Lead SNPs near *SP4*, 8q24, and *IKZF3* reaching genome-wide significance in discovery analyses of CCRLP acute lymphoblastic leukemia patients, including replication analyses in the Children's Oncology Group and the California Childhood Leukemia Study Latino Participants**

| | SP4—rs2390536 risk allele A[a] | | | 8q24.1—rs4617118 risk allele G[b] | | | IKZF3—rs2290400 risk allele T[c] | | |
|---|---|---|---|---|---|---|---|---|---|
| | RAF[d] | P value | OR (95% CI) | RAF[a] | P value | OR (95% CI) | RAF[a] | P value | OR (95% CI) |
| CCRLP meta-analysis[e] | | **$3.59 \times 10^{-8}$** | 1.20 (1.13-1.29) | | **$3.05 \times 10^{-9}$** | 1.27 (1.17-1.38) | | **$2.05 \times 10^{-8}$** | 1.18 (1.11-1.25) |
| CCRLP Latinos | 0.26 | $1.42 \times 10^{-4}$ | 1.19 (1.09-1.31) | 0.15 | $2.90 \times 10^{-5}$ | 1.25 (1.12-1.38) | 0.57 | $4.33 \times 10^{-6}$ | 1.20 (1.11-1.30) |
| CCRLP Whites | 0.35 | $5.69 \times 10^{-5}$ | 1.23 (1.11-1.35) | 0.18 | $1.74 \times 10^{-3}$ | 1.26 (1.09-1.45) | 0.50 | $2.72 \times 10^{-3}$ | 1.15 (1.05-1.26) |
| CCRLP African Americans | 0.22 | 0.73 | 1.07 (0.73-1.58) | 0.23 | $1.55 \times 10^{-3}$ | 1.54 (1.18-2.01) | 0.52 | 0.18 | 1.19 (0.92-1.53) |
| Replication meta-analysis | | 0.075 | 1.10 (0.99-1.23) | | **$1.29 \times 10^{-4}$** | 1.30 (1.14-1.49) | | **0.013** | 1.14 (1.03-1.26) |
| COG Replication (European) | 0.36 | 0.025 | 1.15 (1.018-1.31) | 0.17 | 0.011 | 1.22 (1.05-1.43) | 0.45 | 0.13 | 1.10 (0.97-1.25) |
| CCLS Replication (Hispanic) | 0.2133 | 0.74 | 0.96 (0.77-1.20) | 0.1069 | $1.03 \times 10^{-3}$ | 1.58 (1.20-2.07) | 0.6135 | 0.030 | 1.22 (1.02-1.46) |
| Combined datasets | | **$1.77 \times 10^{-8}$** | 1.18 (1.11-1.24) | | **$1.76 \times 10^{-12}$** | 1.28 (1.19-1.37) | | **$1.06 \times 10^{-9}$** | 1.17 (1.11-1.23) |

RAF risk allele frequency, CCLS California Childhood Leukemia Study, CCRLP California Cancer Records Linkage Project, COG Children's Oncology Group.
Bold values indicate *P* values lower than genome-wide significance ($P < 5 \times 10^{-8}$) for CCRLP GWAS and combined datasets, and nominal significance ($P < 0.05$) for replication meta-analysis.
[a] Rs2390536 was genotyped "on array" for all sample sets apart from COG Replication (Imputed INFO score = 0.99)
[b] rs4617118 was genotyped "on array" for CCRLP datasets, and imputed for COG Replication (INFO = 0.96) and CCLS Replication (INFO = 0.88)
[c] rs2290400 was genotyped "on array" for all sample sets apart from COG replication (INFO = 0.88)
[d] RAF calculated among ethnicity-matched controls
[e] Sample sizes for all analyses: CCRLP Latinos: 1949 cases, 8584 controls; CCRLP whites: 1184 cases, 3551 controls; CCRLP African Americans: 130 cases, 3842 controls. For the replication: COG: 959 cases, 2624 controls; CCLS: 530 cases and 511 controls

Latino controls from the CCLS (a set of cases and controls from California distinct from our discovery set). Using these data, we successfully replicated the novel SNP associations at 8q24 (OR = 1.30, 95% CI: 1.14–1.49, $P_{meta} = 1.29 \times 10^{-4}$) and near *IKZF3* (OR = 1.14, 95% CI: 1.03–1.26, $P_{meta} = 0.013$) (Table 1). For some datasets, the top hit SNP was imputed as shown (Table 1). While the *SP4* locus did not replicate in the combined COG European and CCLS Latinos replication datasets ($P_{meta} = 0.075$), it was nominally associated with ALL risk in the COG sample, with a similar direction and magnitude of effect (OR = 1.15, 95% CI: 1.02–1.31, $P = 0.025$).

SNP imputation enabled improved resolution of the associated regions in discovery samples (Fig. 2a–c). The *SP4* association peak is confined to the *SP4* gene only (Fig. 2a) and conditional analyses adjusting for rs2390536 revealed no independent risk alleles (Supplementary Figure 1). Because rs2390536 did not replicate in our combined replication set of CCLS Latinos and COG Europeans, it may be a false-positive signal in our discovery GWAS and therefore is not explored further here.

The chromosome 8q24 association peak was located within a 2.5 Mb region dense with genome-wide significant loci for a variety of developmental traits, including: cleft lip[14], hypospadias[15], and infant length[16], as well as several cancers[17]. However, our ALL association peak was located within a 100 kb subregion (Fig. 2b), which has not been previously associated with any other phenotype. Conditional analyses adjusting for the lead SNP in the region greatly attenuated SNP associations in the region (Supplementary Figure 2), with the most significant residual association signal found at rs35380634, located some 300 kb telomerically ($P_{conditional} = 3.29 \times 10^{-5}$).

The ALL association peak on chromosome 17q12 is broad, covering approximately 200 kb and encompassing at least six genes, including *IKZF3* (Fig. 2c). The lead SNP, rs2290400, is also a known GWAS hit for type 1 diabetes and asthma[18,19], although the ALL risk allele is protective against development of these other immune-related phenotypes. Conditional analyses adjusting for rs2290400 effectively eliminated any residual association in the region, indicating that additional risk loci proximal to the primary peak likely do not exist (Supplementary Figure 3). Because *IKZF1* also contains ALL risk alleles identified in previous GWAS, we tested for the presence of statistical interaction between our top genotyped SNP in *IKZF1* (rs11978267; $P_{CCRLP} = 2.15 \times 10^{-29}$) and our top genotyped SNP in *IKZF3*. We detected no significant epistatic

effect in either case–control analyses ($P = 0.62$) or case-only analyses ($P = 0.16$).

Our study is designed to discover SNPs associated with case status rather than the clinical course of disease. Within the COG replication dataset, information on patient relapse was available for all 959 cases, allowing for an assessment of risk of relapse. One hundred forty-three patients experienced a relapse, while 816 did not. Logistic regression analyses among these cases did not suggest that either the rs4617118 risk allele on 8q24.1, nor the rs2290400 risk allele near *IKZF3*, was associated with an increased risk of relapse (OR = 1.03 and OR = 0.89; $P = 0.87$ and $P = 0.39$, respectively).

**Functional assessments**. Top hit SNPs in the 8q24 and *IKZF3* region, along with those that displayed larger effect sizes and *P* values within one order of magnitude, were annotated for functional evidence using HaploReg, RegulomeDB, and GTEx (Supplementary Tables 4 and 5). ENCODE-described motifs that bind at SNP locations, RegulomeDB scores, and other GWAS associations are indicated in the tables. Additional expression quantitative trait loci (eQTL) analysis was carried out using RNA sequencing (RNA-seq) data from lymphoblastoid cell lines (LCLs) in the gEUVADIS dataset (www.geuvadis.org).

8q24: The risk locus at 8q24 is located within a gene desert and is most proximal to, but outside of, a noncoding RNA *LINC00977* (Fig. 3). The lead SNP, rs4617118, and those SNPs in tight linkage were not eQTL for any proximal or distal gene in any tissue in GTEx, nor for lymphoblastoid cells in gEUVADIS data. Many GWAS association peaks for other cancers and immune conditions exist on chromosome 8q24. Using chromatin conformation assays on LCLs[20], the region exhibits two large structural chromosomal domains with *MYC* in the center (Fig. 3). SNP rs4617118 exists within one of these domains. This domain shows evidence of chromosome looping between *MYC* and the region containing rs4617118; this region also contains the gene *GSDMC*, on the distal side of rs4617118 (Fig. 3). *GSDMC* is a gasdermin family member, another of which (*GSDMB*) is also present within the chromosome 17q12 peak as explained below.

17q12: The genetic association peak at *17q12* was broad. The top SNP rs2290400 is a highly significant eQTL ($P$ values < $10^{-9}$) for two genes, *GSDMB* and *ORMDL3*, in whole blood (as found on gTEX, Fig. 4). An additional eQTL was identified at *ZPBP2* in the LCL RNA-seq data ($-\log_{10}(P$ value) of 5.737). Structural Hi-C

analysis (using LCL data) did not reveal obvious structural interactions. The variant did not display any evidence of altering expression of *IKZF3* using gTEX or gEUVADIS resources, though the linkage peak clearly covers this gene. Instead, the rs2290400 minor allele, protective against ALL, was clearly associated with *decreased* expression of its other associated gene targets *GSDMB* and *ORMDL3*. Further analysis revealed that rs2290400 disrupts a PRDM1 (BLIMP1) binding motif (Supplementary Table S4).

## Discussion

Using a population-based case-series with both matched controls and additional controls from the same source population (albeit older in age) and two replication datasets, we identified and replicated two new genome-wide significant risk alleles for childhood ALL. These alleles augment our current understanding of the genetic etiology of childhood ALL by (i) adding one haplotype that is a strong eQTL for three genes and located at a well-established hematopoietic developmental gene (*IKZF3*), and (ii) adding an additional locus in a gene-poor region of chromosome 8q24 that is rich with genetic variants affecting risk for myriad different traits. A third association that reached genome-wide significance in our discovery set (at the gene *SP4*) did not replicate in the combined COG and CCLS replication data (*P* = 0.075). Although it is not a confirmed risk factor for childhood ALL, it did reach an overall *P* value below the commonly accepted genome-wide significance level of $5 \times 10^{-8}$ in our meta-analysis and should be assessed in additional datasets.

While the associated haplotype on 17q12 clearly contains *IKZF3*, the lead SNP rs2290400 was not identified as an eQTL for *IKZF3* itself using available databases. Instead, these databases identified the neighboring genes *ORMDL3*, *GSDMB*, and *ZPBP2* as top eQTLs. It is possible that the haplotype imparts more than a single genic effect on risk. Prior GWAS have shown that the ALL risk allele in rs2290400 (T) is protective against asthma[19] and type 1 diabetes[18], suggesting a role for this SNP in immune regulation. An immunologic response role for the 17q12 haplotype is further supported by its association with a variety of other conditions of aberrant immune function, including: primary biliary cirrhosis[21], systemic lupus erythematosus[22], Crohn's disease[23], and ulcerative colitis[24].

It is intriguing that the well-characterized ALL risk allele in *IKZF1* (Ikaros) is also protective against type 1 diabetes[25], paralleling our findings at rs2290400 near *IKZF3*. The seeming paradox wherein a risk allele for asthma and diabetes is associated with *decreased* risk of childhood ALL, a disease also linked to immune activation (as exemplified by an association with documented history of infectious diseases)[26,27], may be explained by complex interactions. While medical diagnosis of infection is associated with an increased risk of childhood ALL, exposure to infectious agents via daycare contact is associated with decreased risk[28]. However, it is important to note that *IKZF3*, *ZPB2*, *ORMDL3*, *GSDMB*, and *GSDMA* are structured in a large haplotype with co-regulatory features[29,30]. The fact that *IKZF3* encodes the TF Aiolos, closely related to another ALL-associated TF Ikaros (*IKZF1*) that is also critical in hematopoietic development and cell fate, and that *IKZF3* has been identified as a recurrent somatic deletion in leukemia[31], suggests that it may be the functional target of our GWAS association here. In support of a causal role for rs2290400, this variant disrupts a binding motif for the B lymphocyte maturation-induced protein 1 (BLIMP1), a TF and known repressor of T-cell activation[32]. It is important to note that BLIMP1 is expressed in early stages of B-cell

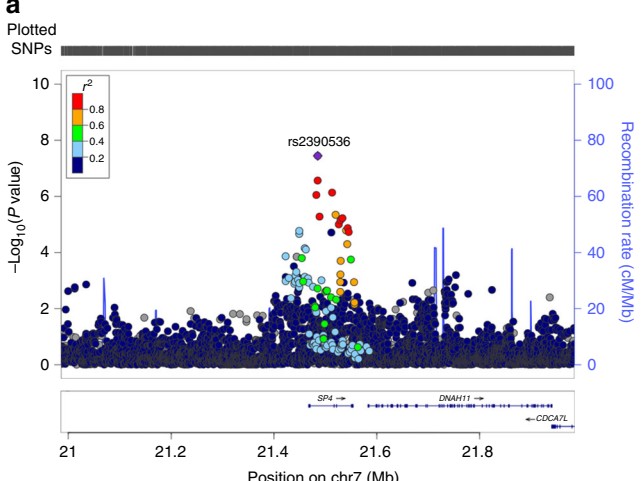

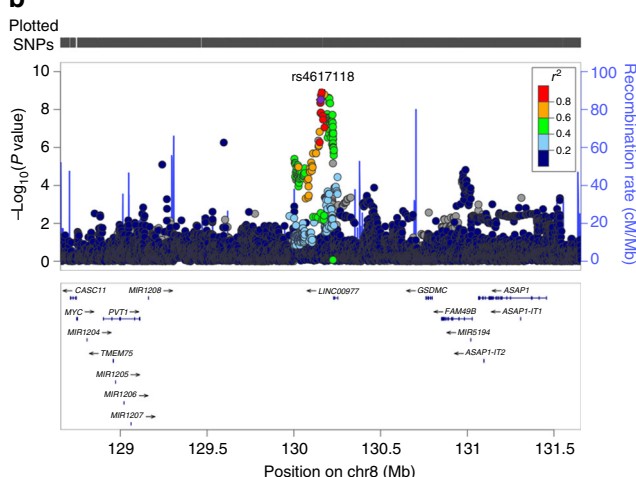

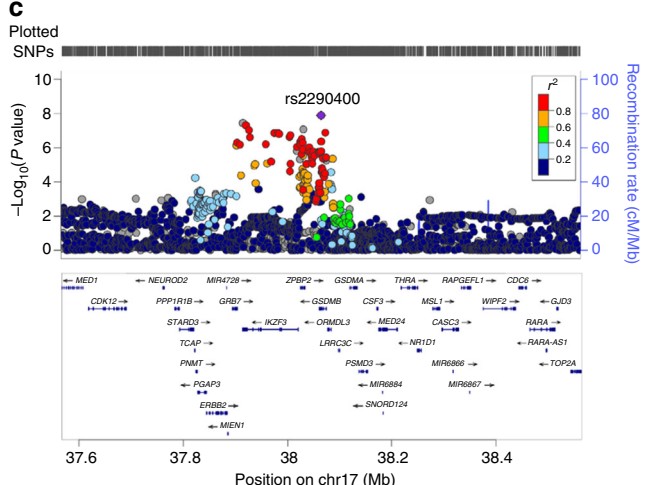

**Fig. 2** Genetic association peaks at chromosomes **a** 7, **b** 8, and **c** 17. Genotypes include both "on array" SNPs and additional SNPs imputed as described in the Methods. The top associated "on array" SNP from the CCRLP/Kaiser discovery analysis is indicated by its "rs" identity (and purple diamond shape), and other SNPs are displayed by color showing their extent of genetic linkage with this top SNP. Recombination rate, genetic position, and the locations of nearby genes are indicated. The *P* values for the association at chromosome 7p15.3 (rs2390536, near *SP4*, OR = 1.20, 95% CI: 1.13–1.29, *P* = $3.6 \times 10^{-8}$), chromosome 8q24.21 (rs4617118, OR = 1.27, 95% CI: 1.17–1.38, *P* = $3.1 \times 10^{-9}$), and chromosome 17q12 (rs2290400, OR = 1.18, 95% CI: 1.11–1.25, *P* = $2.1 \times 10^{-8}$)

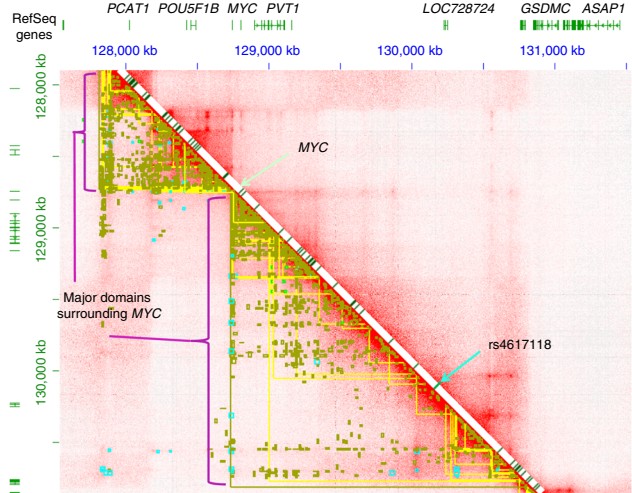

**Fig. 3** Chromatin contacts for the 8q24 top hit SNP. Chromosome contacts for lymphoblastoid cell line GM12878 are shown in the region surrounding the SNP. Two large domains to the left and right of the *MYC* gene are shown (analysis as described by Rao and Huntley[19]), alongside confirmed GWAS catalog alleles for any disease (diagonal line with green hatches indicating previously identified GWAS SNPs for any indication). Despite the large number of disease associations in this region, there are no known diseases with a GWAS hit at the same location as rs4617118. This SNP is located within the genome domain at the right side of the *MYC* locus. On the figure, distant chromatin contacts are indicated in red color off the diagonal, which is the co-occurrence of sequence using the Hi-C method. The annotations below the diagonal line indicate contact points (yellow lines), the top peaks of contact (cyan, or blue color), and chromatin loop calls (green)

development[32], and the SNP may affect risk via aberrant expression control of *IKZF3* at critical hematopoietic differentiation junctures, similar to the activities of other SNPs known to affect B-cell leukemia risk[10]. If the ALL risk allele at rs2290400 favored B-cell differentiation while the alternate allele favored T-cell differentiation, this could potentially account for the inverse association with ALL vs. asthma and type 1 diabetes. We cannot at this point ascertain whether the 17q12 locus exerts functional effects via its impact on hematopoietic differentiation or immune response or both, and hence further research will be required.

The risk locus at 8q24, centered on rs4617118, is a unique GWAS hit in a region rich with genetic associations. It is located 210,000 bases downstream of rs987525, a GWAS SNP for cleft palate[14], and over 400,000 bases upstream of a cluster of SNPs associated with hypospadias, glioma, and white blood cell counts (GWAS catalog). The 8q24 SNP associated with white blood cell counts (rs1991866) is located near *GSDMC*. A related gene on chromosome 17 (*GSDMB*) is found within the leukemia-associated haplotype block containing *IKZF3*. These genes are part of a family of genes involved in pyroptosis, or inflammatory programmed necrosis, which is a recently identified pathway involved in cell proliferation and differentiation, inflammatory remodeling, and cancer-related pathways[33]. Whether any relationship exists between the 8q24 locus and these pathways is unknown. Our preliminary suggestions on function of this allele center around a putative impact on *MYC* regulation, in concert with many other SNPs within the gene desert surrounding both sides of *MYC* which appear to disrupt long-range enhancer activity[34], but do not exclude a potential effect on the function of *GSDMC*, which is harbored within the same looping domain.

Limitations of the current study include a lack of precise immunophenotype and tumor genetic information in our

discovery population. Also, the use of imputed SNPs for some replication analyses as noted in Table 1 may introduce error, although we note that the information scores of our top hit SNPs were high. In addition, our initial goal was to discover aspects of the genetic etiology that explained the higher rate of childhood ALL in Latinos compared to non-Latino whites and the two new loci discovered here do not appreciably account for differences in genetic risk between these populations. Strengths of our discovery population include a population-based sampling strategy for both cases and controls that was not prone to participation bias, and the close match between cases and controls in terms of birth date, sex, and race/ethnicity are evidence for the lack of bias. Our results suggest that genetic risk factors of reasonable strength may continue to be discovered in population-based samples of childhood ALL.

## Methods

**Study subjects.** The study protocol was approved by the Institutional Review Boards at the California Health and Human Services Agency, University of California (San Francisco and Berkeley), and Yale University. The California Department of Public Health, Genetic Diseases Screening Branch obtains blood samples (heel-prick blood spots) from all neonates born within the state for the purpose of genetic screening. The blood spots that remain after genetic screening have been archived at −20 °C since 1982 and made available for appropriate research. We linked statewide birth records maintained by the California Department of Public Health (for the years of 1982–2009) to cancer diagnosis data from the California Cancer Registry (CCR, for the years of 1988–2011). Beginning in January 1988, California law has required all new cancer cases diagnosed in state residents to be reported to the CCR. Included in this analysis were children born in California during 1982–2009 and diagnosed with ALL at the age of 0–14 years per CCR record. Children who were born in California during the same period and not reported to CCR as having any childhood cancer were considered potential controls. For each case of childhood ALL, up to four control subjects were randomly selected from the pool of potential controls and matched to the case on year and month of birth, sex, and race/ethnicity (non-Latino white, non-Latino black, Latino, Asian/Pacific Islander, other). This database of cancer cases and controls is defined as the California Cancer Records Linkage Project (CCRLP). For the current GWAS, a newborn blood spot was pulled from the archive for each case, as well as one control randomly chosen from up to four matched controls for that case. Therefore, active genotyping for CCRLP specifically for this project involved an approximately equal number of cases and controls (see Supplementary Table 1).

**CCRLP sample preparation and genotyping.** DNA was extracted from neonatal blood spots according to the Qiagen DNA Investigator blood card extraction protocol. Three 3 mm punches (Wallac Multipuncher) were cut for each subject and subjected to automated DNA extraction (Qiacube). DNA specimens were assigned randomly to genotyping plates (93 unique samples per plate) and checked for randomization to ensure that case–control status, ethnicity, and sex were distributed evenly. DNA was genotyped on the Affymetrix Axiom World Array (Latino, or LAT) (plates 1–51, 60% of subjects) array[35] and the same array supplemented with MHC and KIR content for immunogenomic imputation purposes[36] on remaining samples (plates 52–86, 40% of subjects). The current manuscript contains analyses of array content common to both array designs, totaling 813,036 variants (i.e., the LAT array base content), plus an additional 8951 variants typed exclusively on plates 52–86. DNA samples were genotyped on an Affymetrix TITAN system, and raw image files were processed with Affymetrix Genetools to call genotypes. Plates 1–51 were clustered separately from plates 52 to 86 due to the differing SNP content, and then called genotypes were merged. Genome freeze hg19 (NCBI 37) was used throughout as the reference.

**CCRLP genotyping quality control.** In addition to randomizing samples, we included two duplicate samples per plate (174 total), with average genotype concordance >99%. SNP filtering was performed separately for the two array designs (i.e., plates 1–51 underwent SNP filtering separately from plates 52 to 86). Call-rate filtering for SNPs and samples were performed iteratively, as follows: SNPs with call rates <92% were removed, then samples with call rates <95% were excluded, then SNPs with call rates <97% were removed, then samples with call rates <96% were excluded. Additionally, 15 CEPH trios (parents and child) were genotyped to detect and exclude SNPs showing >2 Mendelian transmission errors. Any SNP that failed a quality control check on plates 1–51 was also excluded from analysis of plates 52–86, and vice versa. Within the merged dataset of all CCRLP cases and controls, we excluded any SNP displaying significant departure from Hardy–Weinberg equilibrium ($P < 1.0 \times 10^{-4}$) among European-ancestry controls.

In addition to excluding samples due to call-rate filters ($N = 167$), samples with mismatched reported vs. genotyped sex were also excluded ($N = 26$). We performed identity-by-descent analyses in Plink on the merged dataset of all CCRLP cases and

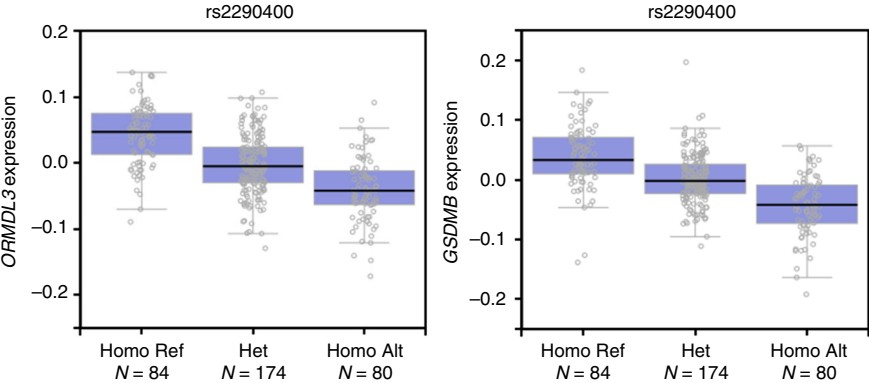

**Fig. 4** GTEX expression quantitative trait analysis of rs2290400 (chromosome 17, reference allele T, alternative allele C). Data from whole blood samples are shown, the significance of the eQTL was lower than a P value of $10^{-25}$ for both genes. Box plots depict the interquartile range, and error bars indicate 1.5 times the interquartile range

controls, stratified by ethnicity, and excluded one member of any sample pair that had an identity-by-decent (IBD) proportion >0.20 (N = 13)[37]. We incorporated genome-wide SNP array data from 1184 HapMap Phase 3 samples and performed multidimensional scaling analysis using unlinked autosomal biallelic SNPs with MAF > 0.05 allowing us to group 89.4% of cases and controls into one of three analytic groups: Latinos, non-Latino whites, and African Americans.

Using hybridization intensity (log R ratio) data from array SNPs on chromosome 21, we identified 109 subjects with Trisomy 21 (~2.9% of total cases) and excluded these subjects from association tests due to the known differences in leukemogenesis in diploid vs. constitutively aneuploid patients[38]. Three additional cases were excluded because, on further examination, it was determined that ALL was not their primary cancer diagnosis.

**Incorporation of additional public controls**. Axiom genotyping data for a total of 73,298 non-Latino white, Latino, and African-American subjects from the Kaiser Resource for Genetic Epidemiology Research on Aging (GERA) Cohort were downloaded from dbGaP (Study Accession: phs000788.v1.p2). All GERA participants are members of Kaiser Northern California Health Care, similar to the CCRLP population. We restricted analyses to only those samples prepared using the Axiom "Type O" Reagent Kit (N = 12,641), as this was the same kit used for CCRLP samples. We excluded any SNPs that were not retained in the cleaned CCRLP dataset, any SNP with an MAF <5%, any SNP associated with control source (CCRLP vs. GERA) at $P < 5 \times 10^{-8}$ in race/ethnicity-stratified analyses, and any SNP displaying significant departure from Hardy–Weinberg equilibrium ($P < 1.0 \times 10^{-4}$) among European-ancestry controls. Samples with call rates <97% were also removed from analyses. The GERA control subject data were then merged with the CCRLP dataset and IBD analyses were performed in Plink, excluding one member of any sample pair that had an IBD proportion >0.20 (N = 161).

**Discovery association analyses**. Following SNP filtering procedures, a total of 757,935 polymorphic autosomal SNPs remained for analyses. Following sample filtering procedures, a total of 3263 cases (1822 males, 1441 females) and 15,977 controls (7704 males, 8273 females) of Latino, non-Latino white, or African-American ethnicity remained for analyses. This included a total of 3506 CCRLP controls and 12,471 GERA controls. Case–control association tests were stratified by ethnicity and included 10533 Latino subjects (1949 cases, 8584 controls), 4735 non-Latino white subjects (1184 cases, 3551 controls), and 3972 African-American subjects (130 cases, 3842 controls). SNP associations were calculated using logistic regression, adjusted for the first 10 ancestry-informative principal components generated by Eigenstrat. Single SNP association results were computed assuming an allelic additive model for 0, 1, or 2 copies of the minor allele. Genomic inflation factors were calculated for each ethnicity-stratified GWAS and suggested no substantial inflation of test statistics ($\lambda_{Latinos} = 1.034$; $\lambda_{non\text{-}Latino\ whites} = 1.004$; $\lambda_{African\ Americans} = 1.000$) following adjustment for the first 10 principal components. The ethnicity-stratified GWAS were subsequently combined using fixed-effects meta-analysis, performed using Plink. SNPs achieving a P value $<5.0 \times 10^{-8}$ in the meta-analysis were considered to have achieved genome-wide statistical significance by current convention and were carried forward for replication analyses[39].

**Replication datasets**. For replication, we used two additional sets of cases and controls. The first is comprised of European-ancestry ALL patients from the COG protocols 9904 and 9905 and controls from the Wellcome Trust Case–Control Consortium. Constitutive DNA from COG samples was extracted from remission blood[40]. DNA samples were genotyped on the Affymetrix 6.0 array and genotype data were downloaded from dbGaP accession phs000638.v1.p1. Genotype data for

European-ancestry control samples genotyped on the Affymetrix 6.0 array were downloaded from the Wellcome Trust Case–Control Consortium[41]. Genotyping quality control procedures were conducted independently in cases and in controls[9]. Briefly, samples with genotyping call rates <98% in cases or controls were excluded. SNPs with genotyping call rates <98% were excluded. We excluded subjects showing evidence of non-European ancestry, samples with mismatched reported vs. genotyped sex, and cryptically related subjects (IBD proportion >0.20), and duplicated subjects already included in our discovery set (N = 21). COG and WTCCC genotype data were merged to create a final set of 959 European-ancestry ALL cases and 2624 controls.

A second replication set consisted of 530 Latino ALL cases and 511 Latino controls from the CCLS that did not overlap with either the CCRLP discovery dataset or the COG replication set. Pre-diagnostic constitutive DNA for CCLS samples was extracted from neonatal bloodcards and genotyped using the Illumina Human OmniExpress V1 platform. DNA extraction was performed using the QIAamp DNA Mini Kit (Qiagen, Valencia, CA, USA). Samples with genotyping call rates <95% were excluded. Samples were screened for cryptic relatedness using unlinked SNPs and excluded if IBD proportion >0.20. Samples with discordant sex information (reported vs. genotyped sex) were excluded. SNPs with genotyping call rates <95% were excluded. Any SNP with a Hardy–Weinberg equilibrium (HWE) P value $<1 \times 10^{-4}$ in controls was excluded.

The COG ALL GWAS dataset were obtained from dbGaP study accession phs000638.v1.p1 (Genome-Wide Association Study of Relapse of Childhood Acute Lymphoblastic Leukemia). The ALL Relapse GWAS dataset was generated at St. Jude Children's Research Hospital and by the COG, supported by NIH grants CA142665, CA21765, CA158568, CA156449, CA36401, CA98543, CA114766, CA140729, and U01GM92666, Jeffrey Pride Foundation, the National Childhood Cancer Foundation, and by ALSAC.

**Replication association analyses and meta-analysis**. Imputation was performed for 1 Mb regions centered on three novel lead SNPs from the discovery meta-analysis. Imputation was performed using the Impute2 v2.1.2 software and its standard Markov chain Monte Carlo algorithm and default settings for targeted imputation[42]. All 1000 Genomes Phase 3 haplotypes were provided as the imputation reference panel[43]. SNPs with imputation quality (info) scores <0.60 or posterior probabilities <0.90 were excluded to remove poorly imputed SNPs. Association statistics for imputed and directly genotyped SNPs were calculated using logistic regression in SNPTESTv2, using an allelic additive model and probabilistic genotype dosages[44]. The effect of individual SNPs on ALL risk was calculated while adjusting for principal components from Eigenstrat.

Association statistics for the COG and CCLS replication datasets were combined using fixed-effects meta-analysis in the META software package to generate a summary odds ratio and P value for the combined replication dataset[45]. The three novel lead SNP associations from discovery analyses were considered to have been successfully replicated if they were statistically significantly associated with ALL risk in this meta-analysis of replication datasets. To preserve the type I error rate, replication P values rejection thresholds were adjusted for multiple comparisons using the Bonferroni–Holm procedure.

A summary meta-analysis of all ALL cases and controls was calculated by combining results from the three ethnicity-stratified discovery samples with the COG and CCLS replication datasets using fixed-effects meta-analysis using the META software[45]. As a measure for between-study heterogeneity, $I^2$ was calculated[46].

**Allelic functional characterization**. Top hit SNPs were associated with genome-wide gene expression using two datasets: the Genotype-Tissue Expression (GTEx)

Project GTEx (Version 6p) on whole blood (n = 338) and LCLs (n = 114)[47], and gEUVADIS RNA sequencing project[48]. The GTEx Project data were obtained from the GTEx Portal on 2 June 2017 and consist of data from dbGAP accession number phs000424. gEUVADIS data is a database of RNA-Seq data derived from 423 LCLs and their accompanying genotype data derived from low-pass whole-genome sequencing as a part of 1000 Genomes project. SNP regions were also assessed with Hi-Seq and Chia-PET chromosomal structure using Juicebox and data from lymphoblastoid cell lines only[49]. Online databases, namely RegulomeDB[50] and Haploreg[51], were utilized to glean functional information on SNPs.

**Data availability**. This study used biospecimens from the California Biobank Program. Any uploading of genomic data and/or sharing of these biospecimens or individual data derived from these biospecimens has been determined to violate the statutory scheme of the California Health and Safety Code Sections 124980(j), 124991(b), (g), (h), and 103850 (a) and (d), which protect the confidential nature of biospecimens and individual data derived from biospecimens. Certain aggregate results may be available from the authors by request.

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

## Acknowledgements

We thank Hong Quach and Diana Quach for DNA isolation support. We thank Martin Kharrazi, Robin Cooley, and Steve Graham of the California Department of Public Health for advice and logistical support. We thank Eunice Wan, Simon Wong, and Pui Yan Kwok at the UCSF Institute of Human Genetics Core for genotyping support. This work was supported by research grants from the National Institutes of Health (R01CA155461, R01CA175737, R01ES009137, P42ES004705, and P01ES018172) and the Environmental Protection Agency (RD83451101), United States. The content is solely the responsibility of the authors and does not necessarily represent the official views of the National Institutes of Health and the EPA. SG is supported by a research grant from the SICPA Foundation. K.M.W. and A.J.d.S. are supported by 'A' Awards from The Alex's Lemonade Stand Foundation. The collection of cancer incidence data used in this study was supported by the California Department of Public Health as part of the statewide cancer reporting program mandated by California Health and Safety Code Section 103885; the National Cancer Institute's Surveillance, Epidemiology and End Results Program under contract HHSN261201000140C awarded to the Cancer Prevention Institute of California, contract HHSN261201000035C awarded to the University of Southern California, and contract HHSN261201000034C awarded to the Public Health Institute; and the Centers for Disease Control and Prevention's National Program of Cancer Registries, under agreement U58DP003862-01 awarded to the California Department of Public Health. The biospecimens and/or data used in this study were obtained from the California Biobank Program, (SIS request #26), Section 6555(b), 17 CCR. The California Department of Public Health is not responsible for the results or conclusions drawn by the authors of this publication. This study makes use of data generated by the Wellcome Trust Case–Control Consortium. A full list of the investigators who contributed to the generation of the data is available from www.wtccc.org.uk. Funding for the project was provided by the Wellcome Trust under award 076113 and 085475. Data came from a grant, the Resource for Genetic Epidemiology Research in Adult Health and Aging (RC2 AG033067; Schaefer and Risch, PIs) awarded to the Kaiser Permanente Research Program on Genes, Environment, and Health (RPGEH) and the UCSF Institute for Human Genetics. The RPGEH was supported by grants from the Robert Wood Johnson Foundation, the Wayne and Gladys Valley Foundation, the Ellison Medical Foundation, Kaiser Permanente Northern California, and the Kaiser Permanente National and Northern California Community Benefit Programs. The RPGEH and the Resource for Genetic Epidemiology Research in Adult Health and Aging are described here: https://divisionofresearch.kaiserpermanente.org/genetics/rpgeh/rpgehhome. For recruitment of subjects enrolled in the CCLS replication set, the authors gratefully acknowledge the clinical investigators at the following collaborating hospitals: University of California Davis Medical Center (Dr. Jonathan Ducore), University of California San Francisco (Drs. Mignon Loh and Katherine Matthay), Children's Hospital of Central California (Dr. Vonda Crouse), Lucile Packard Children's Hospital (Dr. Gary Dahl), Children's Hospital Oakland (Dr. James Feusner), Kaiser Permanente Roseville (formerly Sacramento) (Drs. Kent Jolly and Vincent Kiley), Kaiser Permanente Santa Clara (Drs. Carolyn Russo, Alan Wong, and Denah Taggart), Kaiser Permanente San Francisco (Dr. Kenneth Leung), and Kaiser Permanente Oakland (Drs. Daniel Kronish and Stacy Month). The authors additionally thank the families for their participation in the California Childhood Leukemia Study (formerly known as the Northern California Childhood Leukemia Study).

## Author contributions

J.L.W. and X.M. designed the study, obtained funding, and managed subject selection and recruitment. K.M.W. and A.T.D. performed genotyping quality control and data cleaning, and K.M.W. performed most analyses. A.J.d.S., S.G., S.S.F., J.O., I.S., X.X., L.M., and R.W. performed various analyses. C.M. and R.M.-C. managed CCLS recruitment and analysis. H.M.H. and L.B. managed laboratory sample preparation. All authors contributed to the manuscript preparation which was drafted by J.L.W. and K.M.W.

## Additional information

**Competing interests:** The authors declare no competing financial interests.

