## [Peer Review File · Nature Communications]

Reviewers' comments:

Reviewer #1 Expert in GWAS:

In general the manuscript from Wiemels et. al. presents a well conducted study that provides useful association studies of susceptibility loci for ALL. Notably it includes good support for the identification of a few novel loci. Several aspects of the study should be clarified or revised.

1. Chr 17q12 association.

a. The abstract should indicate that the association at 17q12 is with multiple genes. Although IKZF3 is attractive there is no functional support and arguably the functional data supports other gene(s). The discussion while somewhat balanced I think also needs to be modified. The somatic deletion of IKZF3 in some ALL patients is provocative but it is nearly certain the these deletions included all the other genes in the region i.e. ref 27 does not look at these adjacent genes.

b. The Chr 17q12 association is also found with primary biliary cirrhosis, systemic lupus erythematosus, Crohn's disease and ulcerative colitis. I believe these all have the same "haplotype" but this should be checked.

2. IBD – The study uses a relatively high 0.2 pi hat to exclude related individuals for most of the data sets (line 313, 318, 374). For one data set the study uses an even higher IBD – 0.25 (line 384). It is unclear why this higher threshold was used.

3. Imputation – The investigators should consider genome-wide imputation in the discovery phase rather than just the significant regions. This might identify some additional candidates.

4. The term Latino is used throughout manuscript. It is likely that all or most of the included participants are Mexican American or those from non-Caribbean origin. It would be useful to provide PCA in supplemental information showing the relative admixture of Native American, European and West African ancestry.

5. Residual lambda-gc – I assume that this is after controlling for 10 PCs but this should be clarified. Please also clarify why the SNP effect adjusting for 5 principal components in replication as opposed to 10 PCs in discovery. Perhaps for all analyses adjustments should be for all significant PCs as assessed by Tracy-Widom statistics.

6. Regulome – In the footnote it would be useful to include what each category represents. Some of the categories in Regulome provide only very weak support for function.

7. Supplementary Figures should include a legend.

Reviewer #2 Expert in leukaemia genomics:

Wiemels and Walsh and colleagues have provided a nice analysis on the most common childhood cancer, ALL. They have assessed a large number of both cases and controls for risk alleles, with a primary focus on Hispanics which has been previously understudied. They perform a meta-analysis on the 3 largest ethnic groups in their discovery cohort, and then perform a similar meta-analysis of a European and Hispanic cohort for their validation cohort. They identify 2 new loci which meet statistical criteria for inclusions as ALL risk alleles. These novel observations are very beneficial and would be of great interest to the cancer community. However, justification for the meta-analysis methods and reasoning used is lacking. Potentially additional analyses should be performed prior to acceptance and publication.

1. This analysis has identified 2-3 potentially new loci for ALL. These new hits may have been due to the meta-analyses of both the discovery and validation cohorts using larger sized Hispanic cohort to non-Hispanic cohort ratios than would have been present in past ALL GWAS studies. However, some of the effect estimates are similar across ethnicities which would contradict this reasoning. It would be helpful if the authors could obtain the effect estimates and statistics for these new top loci in previously published ALL studies. This may shed some light on why these observations were not previously observed and strengthen the current findings. If previous studies utilized arrays which did not target these loci and for which imputation is not easily possible, then

a comment on this should be made in the manuscript.

2. Related, the authors used a fixed-effects meta-analysis as opposed to a random-effects meta-analysis. The fixed-effects model assumes there is a single effect measure across all cohorts and is appropriate when combining datasets having a shared ancestry. As the initial premise of the investigation appeared to be to identify novel loci which explain the increase in ALL prevalence in Hispanics, this model is not congruous with such a purpose. A random effects meta-analysis would presume that there is a distribution of effects across ethnic/geographic cohorts. While such a model is likely underpowered with only 2 or 3 groups, it may be more appropriate to that purpose. If the authors changed midstream their purpose from identifying Hispanic alleles to identifying common signals across all cohorts, then a fixed-effects model may be appropriate – and while a subtle statistical change, an explanation of the type of meta-analysis model utilized should be clearly stated as well as this change in purpose. Still, performing a random-effects model meta-analysis and potentially including some of the other less-populated ethnicities may identify additional hits or insights.

3. A fixed effects GWAS meta-analysis of the CCR Hispanics and CCLS Hispanics cohorts may identify Hispanic-specific risk alleles reaching genome-wide significance. The authors are encouraged to perform this exercise on their own if they have not done so already. As there would then be no additional Hispanic replication cohort available, the authors may wish to retain such a finding until they can obtain an additional Hispanic cohort – or including such initial hits in the current paper would certainly strengthen it.

4. Again related, between-study heterogeneity statistics, though calculated as per line 412, are not included nor commented on for any of the loci for either of the 2 meta-analyses performed.

5. In Sup. Table 1, the Meta-Odds Ratio for IKZF1 - rs11978267 is less than the individual odds ratios of each of the constituent cohorts. Please check these numbers as such an occurrence is highly unlikely.

6. Would be great to include the sample sizes of both cases and controls for each of the subgroups in Table 1.

7. CCRLP is used throughout but never defined, only CCR is

8. Would be nice to have consistent use of either “Latino” or “Hispanic” throughout the paper, e.g. Lines 71-72 of the results section contrasted with Table 1.

9. Line 97 $p=5.15 \times 10^{-6}$ is found nowhere in Supplementary Table 1.

10. Might suggest bolding the 5 significant meta-analysis p-values in Table 1 so that the reader can more easily see the reasoning for SP4 being distinct from the other 2 hits as having not validated.

11. A number of typos throughout: Table 1 title “SP1” should be “SP4”, Figure 3 legend “knows” should be “known”, Sup. Table 1 has p-values throughout alternating with “x10” and “X10”, line 138 “cases” should be “case”, line 224, line 288 “LAT” or “Latino”, line 292 and 299 “53” or “52”, line 376 “and” missing.

Reviewer #1 Expert in GWAS:

In general the manuscript from Wiemels et. al. presents a well conducted study that provides useful association studies of susceptibility loci for ALL. Notably it includes good support for the identification of a few novel loci. Several aspects of the study should be clarified or revised.

1. Chr 17q12 association.

a. The abstract should indicate that the association at 17q12 is with multiple genes. Although IKZF3 is attractive there is no functional support and arguably the functional data supports other gene(s). The discussion while somewhat balanced I think also needs to be modified. The somatic deletion of IKZF3 in some ALL patients is provocative but it is nearly certain the these deletions included all the other genes in the region i.e. ref 27 does not look at these adjacent genes.

Response: We agree and therefore have placed the balance of presentation back on to the entire haplotype implicated in the Chr17q12 association. The title was changed to include the chromosome locus only, without specifically mentioning IKZF3, and the abstract contains all of the genes (linked as eQTL) along with IKZF3 in the haplotype in order. The Discussion was also modified to include more consideration on this locus's effects on multiple immune-related disorders and to include the potential SNP effects on either hematopoietic differentiation or general immune regulation. The reviewer's note that the paper by Holmfeldt, et al. does not look at genes adjacent to IKZF3 is of particular interest. Revisiting this manuscript, we observed in their Table S12 that 8% of tumors had disruption of IKZF3 with concomitant loss of the remaining wild-type allele. Three of these mutations were of the entirety of IKZF3 and could certainly encompass neighboring genes. Six of these mutations began in exon 2 of IKZF3 and continued downstream (i.e. away from ZBP2/GSDMB/ORMDL3). The final mutation was a frameshift mutation in IKZF3. While these results do not demonstrate that IKZF3 is the risk gene uncovered by our GWAS, they do suggest that IKZF3 is the primary target of somatic deletions in the region and certainly make it an attractive candidate as the reviewer has stated.

b. The Chr 17q12 association is also found with primary biliary cirrhosis, systemic lupus erythematosus, Crohn's disease and ulcerative colitis. I believe these all have the same "haplotype" but this should be checked.

Response: The haplotype indeed contains SNPs associated with these diseases, and others including atopic march, rheumatoid arthritis, and pediatric autoimmune conditions. The top hit SNP itself (rs2290400) was formally linked in GWAS studies with type 1 diabetes and asthma only, and we would like to concentrate on those for the reverse pleiotropic effects. However, as the reviewer notes, the comparative considerations that we have made with asthma and type 1 diabetes also apply to these other conditions. As an example, rs2872507 is a GWAS hit for ulcerative colitis, Crohn's and rheumatoid arthritis. The 'A' allele is reported to be the risk allele for these three diseases; however, in our data the 'A' allele is protective against development of leukemia ($p=4.2 \times 10^{-5}$). We have added additional references to other immune related conditions (page 10) which helped our Discussion on the potential etiologic role of this risk haplotype.

2. IBD – The study uses a relatively high 0.2 pi hat to exclude related individuals for most of the data sets (line 313, 318, 374). For one data set the study uses an even higher IBD – 0.25 (line 384). It is unclear why this higher threshold was used.

Response: A pi-hat value of 0.20 is frequently used in the literature, although it is on the higher end of typical values. Because the dataset included Californian children and additional Californian controls from Kaiser, our primary concern was excluding first and second degree relatives from analyses. We believed that a threshold of 0.20 would be adequate for this purpose. Additionally, the first estimates of IBD needed to be assessed in the full CCRLP dataset, irrespective of patient ethnicity (since it is quite possible that two Californian children could be second degree relatives, despite belonging to different racial/ethnic groups). IBD estimates tend to be inflated when calculated in cosmopolitan patient sets, thus necessitating our somewhat more conservative approach to pi-hat cutoffs. Regarding the pi-hat cutoff of 0.25 that was applied, we agree that this is too high of a threshold (as a value of 0.24 would certainly indicate a pair of second-degree relatives). We have gone back to these values and can confirm that no subject pairs had pi-hat values >0.20 in that replication dataset. We have edited the manuscript to now reflect that a cutoff of 0.20 was used throughout the analyses.

3. Imputation – The investigators should consider genome-wide imputation in the discovery phase rather than just the significant regions. This might identify some additional candidates.

Response: We agree that whole genome imputation strategy may yield additional new candidates, and we plan on pursuing such an approach in additional analyses outside the scope of the current analysis.

The current manuscript describes those alleles that are genome-wide significant “on array” which have a higher degree of confidence than those discovered via imputation.

4. The term Latino is used throughout manuscript. It is likely that all or most of the included participants are Mexican American or those from non-Caribbean origin. It would be useful to provide PCA in supplemental information showing the relative admixture of Native American, European and West African ancestry.

Response: We have previously shown that Latino leukemia patients recruited in California are primarily of Mexican and Central American backgrounds, with relatively few tracing their ancestry to the Caribbean (e.g. Cuba or Puerto Rico). Prior analyses using the program STRUCTURE indicate that CCLS Latino subject genomes are, on average, 37% Native American, 57% European, and just 6% African (Walsh et al, Leukemia, 2013). Like the CCLS replication subjects, cases and controls appearing in the CCRLP Latino GWAS discovery analysis were California residents at the time of birth (CCRLP cases and controls) or recruitment (additional Kaiser controls). Because we performed PCA analyses with HapMap data, which do not include a Native American reference population, we cannot assign ancestry proportions to the admixed case and control subjects. However, the distribution of admixture is easily inferred from the plot below. As expected, these subjects are broadly distributed across PC1 (representing the Native American/European axis), but show relatively little spread across PC2 (representing the African ancestry/European axis).

5. Residual lambda-gc – I assume that this is after controlling for 10 PCs but this should be clarified. Please also clarify why the SNP effect adjusting for 5 principal components in replication as opposed to 10 PCs in discovery. Perhaps for all analyses adjustments should be for all significant PCs as assessed by Tracy-Widom statistics.

Response: Indeed, the reported lambda values are after adjustment for PCs and this has been clarified in the methods under the section “Discovery association analyses”. Logistic regression analyses of the replication datasets from CCLS and COG were adjusted for the first 5 PCs, consistent with two prior analyses of these datasets that have been published previously (Walsh and de Smith et al. Cancer Research, 2015; Wiemels et al. Leukemia, 2016). As previously reported, the genomic inflation factor for genome-wide analysis of these datasets (after adjusting for the first five PCs) was low (range: $\lambda=1.02-1.05$). We therefore did not believe it necessary to include additional covariates in analyses of these replication datasets. Although the use of Tracy-Widom statistics has been reported to occasionally overestimate the number of significant principal components when analyzing admixed populations (Shriner. Heredity, 2011), we will incorporate this approach into future analyses of these data and appreciate the suggestion.

6. Regulome – In the footnote it would be useful to include what each category represents. Some of the categories in Regulome provide only very weak support for function.

Response: We have now included a footnote for function for each category; indeed, the lower numbered categories hold higher levels of support.

7. Supplementary Figures should include a legend.

Response: We have now included legends for each supplementary figure (below the figure).

Reviewer #2 Expert in leukaemia genomics:

Wiemels and Walsh and colleagues have provided a nice analysis on the most common childhood cancer, ALL. They have assessed a large number of both cases and controls for risk alleles, with a primary focus on Hispanics which has been previously understudied. They perform a meta-analysis on the 3 largest ethnic groups in their discovery cohort, and then perform a similar meta-analysis of a European and Hispanic cohort for their validation cohort. They identify 2 new loci which meet statistical criteria for inclusions as ALL risk alleles. These novel observations are very beneficial and would be of great interest to the cancer community. However, justification for the meta-analysis methods and reasoning used is lacking. Potentially additional analyses should be performed prior to acceptance and publication.

1. This analysis has identified 2-3 potentially new loci for ALL. These new hits may have been due to the meta-analyses of both the discovery and validation cohorts using larger sized Hispanic cohort to non-Hispanic cohort ratios than would have been present in past ALL GWAS studies. However, some of the effect estimates are similar across ethnicities which would contradict this reasoning. It would be helpful if the authors could obtain the effect estimates and statistics for these new top loci in previously published ALL studies. This may shed some light on why these observations were not previously observed and strengthen the current findings. If previous studies utilized arrays which did not target these loci and for which imputation is not easily possible, then a comment on this should be made in the manuscript.

Response: This is a perceptive comment. The original GWAS datasets published for pediatric ALL had 317 cases (Trevino et al) and 577 cases (Papaemmanuil et al) for discovery. These small sample sizes were only able to discover 2-3 loci (ARID5B, CEBPE, IKZF1). We have obtained one of the previously published GWAS datasets here (the COG data, a subset of which was previously described in Trevino, et al.) to use as one of our replication datasets. Despite its limited sample size which would not have enabled discovery of these new risk SNPs, the COG study (of whites) replicated the results found in our discovery dataset for the 8q24 and 17q12 loci with regards to direction and strength of association. The only difference here appears to be the power of discovery afforded by our much larger discovery dataset, although the higher risk allele frequency (RAF) in Hispanics for the 17q12 SNP also likely contributed (i.e. although effect sizes may be similar, differences in allele frequency across populations also influences statistical power).

2. Related, the authors used a fixed-effects meta-analysis as opposed to a random-effects meta-analysis. The fixed-effects model assumes there is a single effect measure across all cohorts and is appropriate when combining datasets having a shared ancestry. As the initial premise of the investigation appeared to be to identify novel loci which explain the increase in ALL prevalence in Hispanics, this model is not congruous with such a purpose. A random effects meta-analysis would presume that there is a distribution of effects across ethnic/geographic cohorts. While such a model is likely underpowered with only 2 or 3 groups, it may be more appropriate to that purpose. If the authors changed midstream their purpose from identifying Hispanic alleles to identifying common signals across all cohorts, then a fixed-effects model may be appropriate – and while a subtle statistical change, an explanation of the type of meta-analysis model utilized should be clearly stated as well as this change in purpose. Still, performing a random-effects model meta-analysis and potentially including some of the other less-populated ethnicities may identify additional hits or insights.

Response: We agree with the reviewer's assessment of the role of fixed and random effect models. The original purpose of this analysis was to discover genetic alleles unique among Latinos with Native American ancestry, in contrast to non-Latino whites who were previously assessed in other GWAS studies. Our analyses did not find alleles with strong heterogeneity between Latinos and Non-Latinos, but did discover new alleles which affected both populations at similar magnitudes. For this reason, we have modified the title and the Introduction to de-emphasize this cross-ethnicity comparison and to instead emphasize the role of new discovery; however it is still an important point that the new discoveries affect both of these populations and, at least for the 8q24 variant, also contributes to ALL risk in African-American children. We do not have sufficient sample size for other less populated ethnicities, such as East Asians, to make additional insights on genetic risk factors in those populations. We have revisited the data to assess how our conclusions may have been affected had we used a random-effects meta-analysis and can report the following. The P-values for the SNPs on chromosomes 7 and 17 (SP4 and IKZF3) would be unchanged, as the random-effects model reduces to the fixed effects model when no heterogeneity is detected ($I^2=0$ for both

these SNPs). In the case of the Chr8 SNP, the discovery p-value would have increased from 3.05×10^{-9} to 1.88×10^{-8} ($I^2=7.2\%$). Based on our threshold to carry SNPs forward to replication ($P < 5.0 \times 10^{-8}$), we would reach the same conclusions had we elected to perform a random-effects meta-analysis of these data. A note about the similarity between random effect and fixed effect model estimates is now included (near the top of page 5).

3. A fixed effects GWAS meta-analysis of the CCR Hispanics and CCLS Hispanics cohorts may identify Hispanic-specific risk alleles reaching genome-wide significance. The authors are encouraged to perform this exercise on their own if they have not done so already. As there would then be no additional Hispanic replication cohort available, the authors may wish to retain such a finding until they can obtain an additional Hispanic cohort – or including such initial hits in the current paper would certainly strengthen it.

Response: This is an intriguing proposal, and as the reviewer intimates, is one that we have planned. As the reviewer points out, we would not have a replication dataset available for Hispanic-specific risk loci, thus such analyses are not yet mature enough to yield convincing results. We have reached out to collaborators at other institutions in Central and South America to follow-up on potential interesting findings in their own patient populations and look forward to these collaborations.

4. Again related, between-study heterogeneity statistics, though calculated as per line 412, are not included nor commented on for any of the loci for either of the 2 meta-analyses performed.

Response: We have modified a version of Table 1 to include I^2 values for study heterogeneity and include this as a supplemental table.

	SP4 - rs2390536 risk allele A			8q24.1 - rs4617118 risk allele G			IKZF3 - rs2290400 risk allele T		
	P-value	OR (95% CI)	I^2	P-value	OR (95% CI)	I^2	P-value	OR (95% CI)	I^2
CCRLP Meta-analysis	3.59×10^{-8}	1.20 (1.13-1.29)	0%	3.05×10^{-9}	1.27 (1.17-1.38)	7.2%	2.05×10^{-8}	1.18 (1.11-1.25)	0%
CCRLP Hispanics	1.42×10^{-4}	1.19 (1.09-1.31)	-	2.90×10^{-5}	1.25 (1.12-1.38)	-	4.33×10^{-6}	1.20 (1.11-1.30)	-
CCRLP Whites	5.69×10^{-5}	1.23 (1.11-1.35)	-	1.74×10^{-3}	1.26 (1.09-1.45)	-	2.72×10^{-3}	1.15 (1.05-1.26)	-
CCRLP African-Americans	0.73	1.07 (0.73-1.58)	-	1.55×10^{-3}	1.54 (1.18-2.01)	-	0.18	1.19 (0.92-1.53)	-
Replication meta-analysis	0.075	1.10 (0.99-1.23)	48.3%	1.29×10^{-4}	1.30 (1.14-1.49)	60.8%	0.013	1.14 (1.03-1.26)	0%
COG Replication (European)	0.025	1.15 (1.018-1.31)	-	0.011	1.22 (1.05-1.43)	-	0.13	1.10 (0.97-1.25)	-
CCLS Replication (Hispanic)	0.74	0.96 (0.77-1.20)	-	1.03×10^{-3}	1.58 (1.20-2.07)	-	0.030	1.22 (1.02-1.46)	-
Combined Datasets	1.77×10^{-8}	1.18 (1.11-1.24)	5.8%	1.76×10^{-12}	1.28 (1.19-1.37)	16.4%	1.06×10^{-9}	1.17 (1.11-1.23)	0%

5. In Sup. Table 1, the Meta-Odds Ratio for IKZF1 - rs11978267 is less than the individual odds ratios of each of the constituent cohorts. Please check these numbers as such an occurrence is highly unlikely.

Response: We thank the reviewer for noticing this – the wrong data for the meta-analysis OR was included; this was now corrected. (note it is 1.43, not 1.24)

6. Would be great to include the sample sizes of both cases and controls for each of the subgroups in Table 1.

Response: We have included this information in a footnote to Table 1, and also a headnote to Supplemental Table 1.

7. CCRLP is used throughout but never defined, only CCR is

Response: We defined CCRLP in the first paragraph of the Methods, where it is first mentioned.

8. Would be nice to have consistent use of either “Latino” or “Hispanic” throughout the paper, e.g. Lines 71-72 of the results section contrasted with Table 1.

Response: We have adopted the term “Latino” as more accurate description of the origin of this population, and therefore have removed the term Hispanic from the tables.

9. Line 97 $p=5.15 \times 10^{-6}$ is found nowhere in Supplementary Table 1.

Response: The GATA3 SNP referenced in the manuscript text, rs570613, was the ‘lead SNP’ from the genotyped data (i.e. the most significantly associated variant on the risk haplotype among SNPs directly typed on-array). In order to maintain consistency with how we discussed results for previously-reported risk SNPs in LHPP and ELK3, we have revised the results section to report on a different GATA3 variant - rs3824662. Like rs35837782 in LHPP and rs4762284 in ELK3, rs3824662 in GATA3 was a previously reported GWAS hit in a prior study, but it was not directly genotyped on our Axiom array. We therefore imputed this

genotype and have replaced data for rs570613 with data corresponding to rs3824662 in Supp Table 1 and in the results section:

“Recently identified GWAS hits at LHPP, ELK3, and GATA3 (rs35837782, rs4762284 and rs3824662, respectively) ^{3,6} were not directly genotyped on the Axiom array, but when those SNPs were imputed in our dataset, replication was achieved in a meta-analysis of the three race/ethnicity-stratified discovery GWAS ($P=5.7 \times 10^{-6}$, 2.1×10^{-3} , and 2.3×10^{-10} , respectively; Supplemental Table 1).”

10. Might suggest bolding the 5 significant meta-analysis p-values in Table 1 so that the reader can more easily see the reasoning for SP4 being distinct from the other 2 hits as having not validated.

Response: We agree and have done this.

11. A number of typos throughout: Table 1 title “SP1” should be “SP4”, Figure 3 legend “knows” should be “known”, Sup. Table 1 has p-values throughout alternating with “x10” and “X10”, line 138 “cases” should be “case”, line 224, line 288 “LAT” or “Latino”, line 292 and 299 “53” or “52”, line 376 “and” missing.

Response: We corrected these and additionally other small issues with the manuscript.

We look forward to your response on our improvements to the manuscript.

Sincerely,

Joseph Wiemels

Reviewers' comments:

Reviewer #1 (Remarks to the Author):

All questions from the original review have been reasonably addressed.

Reviewer #2 (Remarks to the Author):

The authors have revised and improved their original manuscript in several ways. Some items still require additional clarification to enhance readability and insure reproducibility, prior to this manuscript making a nice contribution to childhood ALL research.

1. The source of the controls should be much more clearly explained (and alongside with the cases), e.g. with a clear figure or Table in the supplement depicting the stratification for each ethnicity with its control source (CCR, Kaiser, etc.). The current tallies listed in the abstract of 3,506 matched controls and 12,471 Kaiser controls are not easily deduced from the ethnic control cohort sample sizes. I.e. please provide the sample size for each control source by each ethnicity.

2. The selection of the controls for the cases should be much more clearly described. E.g. the manuscript states that, "up to four control subjects were randomly selected ... and matched to the case on ... race/ethnicity". Yet the ratio of controls to cases is appreciably more than 4 for Latinos and African Americans, and this is likely not reflecting sample exclusion after microarray analysis (ratios in CCRLP cohorts are: Latinos: $8584/1949 = 4.4$, Whites: $3551/1184 = 2.999$, African Americans: $3842/130 = 29.55$).

3. Much of the control data in the discovery cohorts came from Kaiser's GERA cohort. Were the authors able to restrict selection of these controls by geographic location (i.e. to California only)? If so this should be mentioned explicitly. If not, comment on this should be made including the approximate percent of the samples which would likely be attributed to having a California origin. Line 188 is not sufficiently descriptive if so. As Kaiser covers several states, this detail would help the readership understand better the amount of geographic control utilized in this study. If one of the cohorts was more biased to a particular region of California, this could be briefly mentioned.

4. The replication cohorts utilized Affy SNP6 and Illumina OmniExpress chips as opposed to the Axiom array utilized in the discovery cohort. Lines 102-111 of the manuscript are not clear as to whether the lead SNPs were actually on all 3 microarrays, or whether for some imputation was necessary. This should be commented on, and if any of the SNPs in Table 1 were imputed for any of the cohorts, a note of that for each occurrence should also be made in that table (such as is done in ST1) and mentioned as a potential limitation of the study.

5. Legends for Figures 2A-C should be labeled as showing p-values for the meta-analysis of the discovery cohort.

6. Would be helpful to provide a color scale and definition for figure 3, e.g. turquoise, gold, red defined.

7. The sizes of the replication cohorts in Table 1 have not been provided there as was provided in the revision for the discovery cohorts.

8. Please describe more precisely the block randomization utilized and mentioned on lines 294-295. I.e. were equal proportions of cases and controls and the other covariates present on the two sets of plates, with the different array types being the only blocking variable?

9. Lines 419-421 mentions that the "replication p-values were adjusted for multiple comparisons"

by the Bonferroni-Holm procedure. If the p-values listed in Table 1 are adjusted p-values, a comment in a footnote there should be provided. If the p-values listed are unadjusted and instead only the procedure was followed for hypothesis testing, then lines 419-421 should be modified to state that, e.g. "replication p-value rejection thresholds were adjusted for multiple comparisons ...".

Minor:

Nice if ST3 can be resized to fit on only a single page (as opposed to current drifting of some text onto a second page)

Typos: "gVEUDIS"; "shows" in figure 3 legend

Use of hg19 should likely be mentioned in the methods if so

Reviewers' comments:

Reviewer #1 (Remarks to the Author):

All questions from the original review have been reasonably addressed.

Response: Thank you.

Reviewer #2 (Remarks to the Author):

The authors have revised and improved their original manuscript in several ways. Some items still require additional clarification to enhance readability and insure reproducibility, prior to this manuscript making a nice contribution to childhood ALL research.

1. The source of the controls should be much more clearly explained (and alongside with the cases), e.g. with a clear figure or Table in the supplement depicting the stratification for each ethnicity with its control source (CCR, Kaiser, etc.). The current tallies listed in the abstract of 3,506 matched controls and 12,471 Kaiser controls are not easily deduced from the ethnic control cohort sample sizes. I.e. please provide the sample size for each control source by each ethnicity.

Response: Agreed; there was nowhere in the manuscript where CCRLP controls were enumerated separately from Kaiser controls by ethnicity. We added Supplementary Table 1 to delineate the sample sets more clearly. This table is now referred to in both the Results and in the Methods.

2. The selection of the controls for the cases should be much more clearly described. E.g. the manuscript states that, “up to four control subjects were randomly selected ... and matched to the case on ... race/ethnicity”. Yet the ratio of controls to cases is appreciably more than 4 for Latinos and African Americans, and this is likely not reflecting sample exclusion after microarray analysis (ratios in CCRLP cohorts are: Latinos: $8584/1949 = 4.4$, Whites: $3551/1184 = 2.999$, African Americans: $3842/130 = 29.55$).

Response: The CCRLP was set up for both epidemiologic (data only) analysis and the genetic analysis described in this manuscript. For the genetic analysis, we only used one case per each control to control costs, and because we were also able to access additional controls via the Kaiser GERA cohort. One out of the four available controls was chosen for each case. (this was stated at the bottom of page 13 in the prior submission, “For the current GWAS, a newborn blood spot was pulled from the archive for each case, as well as one control randomly chosen from up to four matched controls for that case.” When adding the additional available controls from GERA, the ratio of controls to cases was greatly increased. We agree that this was not explained well and have made changes to the manuscript in the

Results, and added Supplementary Table 1.

3. Much of the control data in the discovery cohorts came from Kaiser's GERA cohort. Were the authors able to restrict selection of these controls by geographic location (i.e. to California only)? If so this should be mentioned explicitly. If not, comment on this should be made including the approximate percent of the samples which would likely be attributed to having a California origin. Line 188 is not sufficiently descriptive if so. As Kaiser covers several states, this detail would help the readership understand better the amount of geographic control utilized in this study. If one of the cohorts was more biased to a particular region of California, this could be briefly mentioned.

Response: Kaiser does indeed cover several states, but the GERA study is confined to participants in the Kaiser Department of Research region covering Northern California. All GERA participants are long time Kaiser members residing in Northern California. We have added this information to the manuscript.

4. The replication cohorts utilized Affy SNP6 and Illumina OmniExpress chips as opposed to the Axiom array utilized in the discovery cohort. Lines 102-111 of the manuscript are not clear as to whether the lead SNPs were actually on all 3 microarrays, or whether for some imputation was necessary. This should be commented on, and if any of the SNPs in Table 1 were imputed for any of the cohorts, a note of that for each occurrence should also be made in that table (such as is done in ST1) and mentioned as a potential limitation of the study.

Response: We show this information below, and have added this information to Table 1 as well. In addition, we made a note of this potential limitation in the Discussion.

- **rs2390536 (SP4): on-array for CCRLP; on-array for CCLS; imputed for COG (INFO=0.99)**
- **rs4617118 (Chr8q24): on-array for CCRLP; imputed for CCLS (INFO=0.88); imputed for COG (INFO=0.96)**
- **rs2290400 (IKZF3): on-array for CCRLP; on-array for CCLS; imputed for COG (INFO=0.88)**

5. Legends for Figures 2A-C should be labeled as showing p-values for the meta-analysis of the discovery cohort.

Response: Agreed and done.

6. Would be helpful to provide a color scale and definition for figure 3, e.g. turquoise, gold, red defined.

Response: description for these annotations are now included.

7. The sizes of the replication cohorts in Table 1 have not been provided there as was provided in the revision for the discovery cohorts.

Response: we have added the replication cohort numbers to the footnotes. Thank you for noticing the omission.

8. Please describe more precisely the block randomization utilized and mentioned on lines 294-295. I.e. were equal proportions of cases and controls and the other covariates present on the two sets of plates, with the different array types being the only blocking variable?

Our samples were randomized, grouped into sets of 93 unique samples (per plate, excluding the two duplicates and one control DNA sample positioned in a unique location for each plate), and checked to make sure that case/control status, ethnicity, and gender was adequately random on all the plates. Technically this is not “block randomization” since we did not block on the attributes as part of the randomization process, we only checked afterwards. We made this more clear in the manuscript.

There were 2 different array types run, the second having had additional content for HLA and KIR genotyping. Apart from those SNPs, the arrays were the same. The samples arrayed on the first and second array type are part of the same randomized list of specimens.

9. Lines 419-421 mentions that the “replication p-values were adjusted for multiple comparisons” by the Bonferroni-Holm procedure. If the p-values listed in Table 1 are adjusted p-values, a comment in a footnote there should be provided. If the p-values listed are unadjusted and instead only the procedure was followed for hypothesis testing, then lines 419-421 should be modified to state that, e.g. “replication p-value rejection thresholds were adjusted for multiple comparisons ...”.

Response: The latter is true and we have changed the text to reflect this.

Minor:

Nice if ST3 can be resized to fit on only a single page (as opposed to current drifting of some text onto a second page)

Response: it is difficult to resize as such without splitting into more tables; we used Page Setup to permit landscape printing limited to 1 page (43% sized)

Typos: “gVEUDIS”; “shows” in figure 3 legend

Use of hg19 should likely be mentioned in the methods if so

Response: we have done these.

Sincerely,

Joseph Wiemels on behalf of all the authors.

REVIEWERS' COMMENTS:

Reviewer #2 (Remarks to the Author):

Thank you for clearing up those last requested aspects. The paper is now more clear and reads very well. Good work!